# Cutaneous Squamous Cell Carcinoma: From Pathophysiology to Novel Therapeutic Approaches

**DOI:** 10.3390/biomedicines9020171

**Published:** 2021-02-09

**Authors:** Luca Fania, Dario Didona, Francesca Romana Di Pietro, Sofia Verkhovskaia, Roberto Morese, Giovanni Paolino, Michele Donati, Francesca Ricci, Valeria Coco, Francesco Ricci, Eleonora Candi, Damiano Abeni, Elena Dellambra

**Affiliations:** 1IDI-IRCCS, Dermatological Research Hospital, via di Monti di Creta 104, 00167 Rome, Italy; f.dipietro@idi.it (F.R.D.P.); s.verkhovskaia@idi.it (S.V.); r.morese@idi.it (R.M.); francesca.ricci@idi.it (F.R.); fraric1984@gmail.com (F.R.); candi@uniroma2.it (E.C.); d.abeni@idi.it (D.A.); e.dellambra@idi.it (E.D.); 2Department of Dermatology and Allergology, Philipps University, 35043 Marburg, Germany; didona@med.uni-marburg.de; 3Unit of Dermatology, IRCCS Ospedale San Raffaele, 20132 Milano, Italy; paolino.giovanni@hsr.it; 4Department of Pathology, University Hospital Campus Bio-Medico, 00128 Rome, Italy; micheledonati25@gmail.com; 5Sikl’s Department of Pathology, Medical Faculty in Pilsen, Charles University in Prague, 30166 Pilsen, Czech Republic; 6Institute of Dermatology, A. Gemelli University Polyclinic, IRCCS and Foundation, Sacred Heart Catholic University, 00168 Rome, Italy; cocovaleria@hotmail.it; 7Department of Experimental Medicine, University of Rome Tor Vergata, Via Montpellier 1, 00133 Rome, Italy

**Keywords:** squamous cell carcinoma, non-melanoma skin cancer, keratinocyte carcinoma, dermoscopy, therapy, radiotherapy, immunotherapy, Bowen’s disease, cemiplimab

## Abstract

Cutaneous squamous cell carcinoma (cSCC), a non-melanoma skin cancer, is a keratinocyte carcinoma representing one of the most common cancers with an increasing incidence. cSCC could be in situ (e.g., Bowen’s disease) or an invasive form. A significant cSCC risk factor is advanced age, together with cumulative sun exposure, fair skin, prolonged immunosuppression, and previous skin cancer diagnoses. Although most cSCCs can be treated by surgery, a fraction of them recur and metastasize, leading to death. cSCC could arise de novo or be the result of a progression of the actinic keratosis, an in situ carcinoma. The multistage process of cSCC development and progression is characterized by mutations in the genes involved in epidermal homeostasis and by several alterations, such as epigenetic modifications, viral infections, or microenvironmental changes. Thus, cSCC development is a gradual process with several histological- and pathological-defined stages. Dermoscopy and reflectance confocal microscopy enhanced the diagnostic accuracy of cSCC. Surgical excision is the first-line treatment for invasive cSCC. Moreover, radiotherapy may be considered as a primary treatment in patients not candidates for surgery. Extensive studies of cSCC pathogenic mechanisms identified several pharmaceutical targets and allowed the development of new systemic therapies, including immunotherapy with immune checkpoint inhibitors, such as Cemiplimab, and epidermal growth factor receptor inhibitors for metastatic and locally advanced cSCC. Furthermore, the implementation of prevention measures has been useful in patient management.

## 1. Introduction

Cutaneous squamous cell carcinoma (cSCC) is the second most common nonmelanoma skin cancer (NMSC) after basal cell carcinoma (BCC). cSCC accounts for 20% of cutaneous malignancies and about 75% of all deaths due to skin cancer, excluding melanoma. Its incidence rate is constantly increasing, mainly due to population aging and the focus on skin cancer screening [1].

cSCC originates from an uncontrolled proliferation of atypical epidermal keratinocytes, and it is probably the result of a long-lasting intraepidermal dysplasia process. Indeed, tumor development is known to be a gradual process with several histological and pathological defined stages along its malignant evolution from actinic keratosis (AK) to invasive cSCC [2]. Although rarely metastatic, cSCC can favor local cutaneous destruction involving also soft tissue, cartilage, and bone. Generally, the prognosis of cSCC is good and the five-year survival is ≥90% [3,4]. Risk factors involved in the etiopathogenesis are mainly ultraviolet radiation exposure, chronic photoaging, age, male sex, immunosuppression, smoking, and specific genetic factors [4,5].

Several histopathological cSCC forms have been described with different prognostic values. Dermoscopy and, more recently, reflectance confocal microscopy enhanced the diagnostic accuracy of cSCC. Although the surgical approach is the first-line treatment for invasive cSCC, other techniques (i.e., curettage, electrodessication, cryosurgery, lasers, and photodynamic therapy) are available for noninvasive forms. Surgical excision alone guarantees a good prognosis, with cure rates greater than 90% [6]. In patients not candidates for surgery (i.e., locally advanced disease), radiotherapy (RT) may be considered as a primary treatment. 

Extensive studies on cSCC pathogenic mechanisms identified several pharmaceutical targets. Indeed, the multistage process that leads to cSCC development and progression is characterized by mutations in the genes involved in epidermal homeostasis and by several alterations, such as epigenetic modifications, viral infections, or microenvironmental changes [2,7]. Thus, the identification of several drug targets allowed the development of new systemic therapies, including immunotherapy, epidermal growth factor receptor (EGFR) inhibitors, chemotherapy, and electrochemotherapy [8]. For instance, the immune checkpoint inhibitor cemiplimab is approved in the USA and Europe as a systemic treatment for metastatic and locally advanced cSCC not amenable to surgical or RT treatment. Adjuvant therapy consists of an additional treatment, systemic or RT, added to reduce the risk of recurrence after complete surgical excision of the tumor [8].

This review presents a literature overview of cSCC from pathophysiology to novel therapeutic approaches.

## 2. Epidemiology

A 2007 review on the epidemiology of NMSC stated that “accurate and comparable data are needed” to describe the occurrence of these cancers in humans [9]. However, not much progress has occurred in the registration of these frequent malignancies, so it would be honest to admit that the true incidence of cSCC is largely unknown. In fact, most estimates have wide margins of variability, and such heterogeneity is further emphasized by vast geographical differences. In Australia, for instance, estimates on the incidence of cSCC were as high as 499 per 100,000 among males and 291 per 100,000 among females [10]. In Europe, while the point estimates are much lower, the heterogeneity of the incidences is quite staggering: In different studies, the incidence of cSCC ranged from 9 to 96 per 100,000 among males and from 5 to 68 per 100,000 among females [11,12,13]. To further complicate this sketchy picture, it has to be considered that all these figures most likely represent a vast underestimate of cSCC incidence, as they are based on pathology reports while many cSCC are now treated with strategies that do not include excision of the tumor.

## 3. Risk Factors

The etiology of cSCC is multifactorial and includes environmental, immunological, and genetic factors. Environmental factors primarily include cumulative ultraviolet radiation (UVR) exposure (both sun exposure and tanning devices) [4,14]. cSCC is prevalent in the elderly population (80% occurs in people over 60 years old) and more frequent among men than women, given the association with cumulative sun exposure (including professional and leisure-time exposures). Fair complexion is another of the main risk factors [15,16]. cSCC risk is mostly associated with cumulative lifetime sun exposure, while intermittent and intense sun exposure increases the risk of BCC [17]. The use of tanning beds, especially early in life (<25 years), results in an increased risk of cSCC [18]. Furthermore, patients who have had psoralen and ultraviolet A (PUVA) treatment for skin diseases may also be at higher risk [19]. Other risk factors are beta-human papilloma virus (HPV) subtypes infection [20,21], smoking [22,23], and immunosuppression. Higher rates of cSCC have been observed in organ transplantation recipients [24], in persons with human immunodeficiency virus (HIV) infection [25], and in hematopoietic stem cell transplanted patients [26]. The duration of immunosuppression may contribute to cSCC carcinogenesis [27]. Invasive cSCC may develop ex novo, or from a preexisting in situ cSCC (AK or Bowen’s disease, BD), and/or in the context of some chronic photo-exposed or inflammation skin areas. Long-term cutaneous inflammation—such as that observed in chronic wounds, burns, scars, ulcers, or sinus tracts—seems to contribute to cSCC development [16]. Pharmacological treatments with BRAF inhibitor monotherapy (i.e., vemurafenib, dabrafenib, or encorafenib) have a higher risk of cSCC development [28] compared to combined BRAF/mitogen-activated protein kinase kinase (MEK) inhibitors [29]. Development of cSCC during vismodegib (a Hedgehog pathway inhibitor) treatment and voriconazole has also been reported [30,31]. The use of photosensitizing thiazide antihypertensives and cSCC development has been debated. Although the meta-analysis by Gandini et al. [32] reported no association between this treatment and cSCC, a possible link has been described in the meta-analysis by Tang et al. [33]. Furthermore, previous history of cSCC represents a risk factor for additional skin cancers, including other NMSC and melanoma. Flohil et al. showed that the proportion of a subsequent cSCC, BCC, or melanoma in cSCC patients was respectively 13.3%, 15.9%, and 0.5% (0.3–0.6%) [34]. A recent study found that NMSC patients had a relative risk (RR) for melanoma of 6.2 compared to controls, and melanoma risk was particularly high in patients who had NMSC before the age of 40 (RR 25.1) [35]. Finally, some genetic conditions may predispose to cSCC development, such as recessive dystrophic epidermolysis bullosa, albinism, xeroderma pigmentosum, Fanconi anemia, and Lynch syndrome/Muir–Torre syndrome [36]. 

## 4. Clinical Features

The clinical features of cSCC are extremely polymorphous, and they depend also on the anatomical site and subtype. 

### 4.1. Bowen’s Disease (BD)

BD, also known as in situ cSCC, is mostly characterized by a red, sharply demarcated, scaly plaque. It is most frequently detected on sun-exposed skin of the head, neck, and extremities (Figure 1A). Up to 5% of in situ cSCC may evolve into invasive cSCC [37]. Erythroplasia of Queyrat is a particular form of in situ cSCC that arises on the penis. It is characterized by a velvety erythematous lesion [38].

### 4.2. Keratoacanthoma (KA)

KA can be also considered as a subtype of cSCC. It is more frequently detected in Caucasian males in their 60s (Figure 1B). KA is also related to ultraviolet (UV) irradiation, HPV exposure, immunodeficiency, and DNA repair anomalies [39,40]. KA can also develop from scars, trauma, and laser resurfacing [38,39]. Furthermore, the development of multiple KA is described in patients affected by inherited syndromes, such as Muir–Torre syndrome and Witten–Zak syndrome [38]. Its main clinical feature is represented by a spontaneous regression after a rapid growth [38]. 

### 4.3. Invasive cSCC

Invasive cSCC is mostly detected on sun-exposed skin and generally present as a persistent ulcer (Figure 1C) or nonhealing wound [38]. It has been reported that up to 55% of all cSCC develops on the head and neck, while the dorsal area of the hands and forearms are involved in up to 18% of cases [38]. Furthermore, legs, back, and upper extremities are respectively involved in 13%, 4%, and 3% of the cases [38]. However, cSCC can involve any anatomical area, including the lips, anus, and genitals [38]. Clinical features of cSCC depend largely on the degree of differentiation of the lesion. On the one hand, well differentiated cSCC manifests as scaly nodes ore plaques; on the other hand, poorly differentiated cSCC presents mostly as soft, ulcerated, or hemorrhagic lesions. Marjolin’s ulcer is a particular type of cSCC that develops on a burn wound, usually on the lower extremities. Marjolin’s ulcer shows in up to 32% of cases lymph node metastases at the time of the diagnosis [41]. Furthermore, distant metastases have been reported in 27% of cases [41]. HPV-related cSCC commonly presents as a new or enlarging wart on genital and periungual areas [41]. Usually, patients reported a history of refractory warts.

Depending on the extension, cSCC can be classified as common primary or advanced cSCC. Common primary cSCCs are nonmetastatic cSCCs that can be easily removed surgically; they can be further classified as low-risk or high-risk cSCC, depending on the risk of recurrence [8,14]. High-risk factors include several clinical and pathological variables, such as size and location of lesion, poorly defined borders, and rapidly growing tumor [8,14,42]. More details on high-risk factors are reported in Table 1.

## 5. Histology

Several subtypes of cSCC are described [40]. cSCC can be associated with metastatic disease. The main metastatic sites are the lymph nodes, but lung, bone, brain, and mediastinum may also be affected [40]. Several histopathological parameters may be associated with an increased risk of metastasis, such as a deeper skin invasion, lesions greater than 2 cm in diameter, and perineural invasion [40]. For these reasons, an accurate histopathological diagnosis plays a pivotal role in the management of this malignancy.

### 5.1. BD

BD is an intraepidermal carcinoma with atypia of keratinocytes at all levels of the epidermis (in situ cSCC) [43]. The main histological features are parakeratosis, hyperkeratosis with an epidermis characterized by the presence of disordered maturation with atypical keratinocytes through all the epidermal layers, individual cell keratinization, pleomorphism of nuclei, atypical mitoses, and multinucleated tumor cells (Figure 2A). The basal layer is often not altered [43]. Keratinocytes can show pagetoid changes, while in other cases they show extensive clear cell changes. In the papillary dermis, a mixed inflammatory infiltrate characterized by lymphocytes and plasma cells can be often detected [43]. To exclude a melanoma, S100, Human Melanoma Black (HMB)-45 and Melan-A, and cytokeratin stains are usually employed [43].

### 5.2. Keratoacanthoma (KA)

KA is a symmetrical and circumscribed proliferation of keratinocytes with central horn plug and epidermis that extends over the tumor. KA can be classified also as a highly differentiated cSCC [43]. It is characterized by rapid growth and a tendency to a spontaneous resolution. Although some authors consider keratoacanthoma as a unique entity, other authors consider KA as a variant of cSCC [43].

### 5.3. Invasive cSCC

Invasive cSCC is an epithelial malignancy characterized by atypical keratinocytes with a locally destructive growth and increased risk of metastatization. Histologically, it is mainly characterized by atypical and dyskeratotic keratinocytes with hyperchromatic and pleomorphic nuclei with mitoses (Figure 2B). Well-differentiated cSCCs typically have horn pearls and single cell keratinization, while poorly differentiated cSCCs usually show a lack of keratinization and many atypical mitoses. A mixed inflammatory infiltrate is usually present. Since the grade of differentiation is an important predictor, Broder’s staging is used to assess this parameter: grade I includes tumors composed of <25% undifferentiated cells, grade II lesions with <50% undifferentiated cells, grade III lesions with <75% undifferentiated cells, and grade IV lesions with >75% undifferentiated cells [40]. 

### 5.4. Other cSCC Variants

In addition to the classical variant described above, eight histological variants can be detected: desmoplastic, spindle-cell, cantholytic, pseudovascular, verrucous, epithelioma cuniculatum, adenosquamous, and neurotropic cSCC [40].

Desmoplastic cSCC is characterized by a desmoplastic stroma that involves at least 30% of the stroma. It is mainly associated with a metastatic spread [40,43].

Spindle-cell cSCC often arises on post-traumatic scars. Histologically, it is characterized by spindle cells that involve the dermis, but with a stromal desmoplasia less than 30%, the stroma is usually myxoid with pleomorphic cells [40].

Acantholitic cSCC (adenoacanthoma) usually arises in the head and neck region in elderly individuals. Histologically, it shows an adenoid pattern with several dyskeratotic cells with a single floating cell. The negativity to cantholyticnic antigen and positivity for epithelial membrane antigens allow differentiation of cantholytic cSCC from eccrine neoplasms [40].

The pseudovascular cantholytic (or adenoid) cSCC is characterized by the presence of a prominent glandular appearance with pseudovascular structures and malignant cells with hobnail aspects that dissect collagen bundles [40]. 

Verrucous cSCC (also known in genital areas as Buschke–Löwenstein tumor) is characterized by exophytic squamous proliferation with marked papillomatosis and low atypia, the presence of koilocyte-like changes, and a central collection of neutrophils (Figure 2C). 

Epithelioma cuniculatum often arises on the foot; sometimes it is preceded by a plantar wart. Histologically, it shows hyperkeratosis, acanthosis with an undulating, densely keratinized, well differentiated squamous epithelium [40]. 

Adenosquamous cSCC is another type of cSCC characterized by a mixed glandular and squamous differentiation with an aggressive behavior (Figure 3A) [40]. 

Finally, neurotropic cSCC is characterized by a perineural spread (Figure 3B), while papillary cSCC often arises in elderly women and in immunosuppressed patients and is characterized by a prominent papillary growth pattern, usually without deep invasion (Figure 3C) [40].

## 6. Dermoscopy

Dermoscopy can be helpful to differentiate the various forms of cSCC, including pigmented or nonpigmented BD, and invasive cSCC (Table 2).

### 6.1. BD

Dotted and glomerular vessels and scaly white-to-yellow surfaces are commonly observed in non-pigmented BD (Figure 4A) [44,45,46]. Otherwise, brown to gray globules/dots, and structureless pigmentation were observed in 21–80% and 70–78% of pigmented BD, respectively (Figure 4B) [46,47,48]. It has been reported that the pigmented globules/dots are generally distributed at the periphery, sometimes exhibiting a streak-like or leaflike structure. Nevertheless, they are not specific for pigmented BD. Two new dermoscopic features have been recently described: parallel pigmented edges at the periphery of the lesion, named the double-edge sign; and several aggregated large pigmented massive structures, generally at the periphery of the lesion, called clusters of brown structureless areas [49].

### 6.2. Invasive cSCC

The presence of keratin/scales, blood spots, white circles, white structureless areas, hairpin vessels, linear-irregular vessels, perivascular white halos, and ulceration are the main dermoscopic features of invasive cSCC (Figure 4C,D) [50,51,52]. Keratin/scales are homogeneous opaque yellow to brown structures due to hyperkeratosis and parakeratosis [51,53]. Blood spots consist of multiple red to black dots in the keratin mass, corresponding to small crusts or hemangiomas [51]. White circles consist of bright white circles surrounding a dilated infundibulum, corresponding to acanthosis and hypergranulosis of the infundibular epidermis [53]. White structureless areas consist of whitish areas covering large areas of tumors, corresponding to large targetoid hair follicles [54].

Considering the abovementioned dermoscopic features, keratin/scales are strong predictors of well and moderately differentiated cSCC, while the presence of vessels in more than half of the tumor’s surface with a diffuse distribution of vessels and bleeding is a predictor of poorly differentiated cSCC [55]. Furthermore, keratin and white circles resulted in a 79% and 87% sensitivity and specificity for cSCC diagnosis, respectively [51]. 

## 7. Reflectance Confocal Microscopy

Reflectance confocal microscopy (RCM) represents an add-on tool for the noninvasive diagnosis and management of cSCC. The key RCM features of SCC are the presence of an atypical honeycomb pattern or disarranged pattern of the spinous-granular layer, round nucleated cells at the spinous-granular layer, and round blood vessels traversing through the dermal papillae perpendicular to the skin surface (Figure 5) [56,57]. The scale crust appears as brightly reflective amorphous islands on the surface of the skin. Polygonal nucleated cells at the stratum corneum represent parakeratosis, while round nucleated cells in the spinous-granular layer correspond to dyskeratotic cells. Compared to AKs, the spinous-granular layer in cSCC showed more extensive atypia [56]. Furthermore, in cSCC a severe architectural disarray is present in the stratum granulosum, and the number of blood vessels and diameter of the vessels is increased [57,58,59]. This latter feature can be explained by the higher metabolic requirement of the tumor [60,61]. Peppelman et al. demonstrated that the presence of architectural disarray in the stratum granulosum in combination with architectural disarray in the stratum spinosum and/or nest-like structures were the best predictors for cSCC [59]. Recent studies reported that RCM image analysis performed by expert readers achieved sensitivity values ranging from 80.0% to 93.3% and specificity values ranging from 88.3% to 98.6% [54]. For invasive cSCC, large and comprehensive RCM studies are still lacking. Small-sized studies showed that the main RCM pattern of invasive cSCC is a disarranged or atypical honeycomb pattern in the epidermis, round nucleated bright cells in the supra-basal epidermis, and looping blood vessels in dermal papillae [62].

## 8. Pathogenesis of cSCC

A complex network of deregulated signaling pathways plays an important role in the pathogenesis of cSCC. AK and cSCC lesions result from a multistage process, involving mutations in the genes implicated in epidermal homeostasis promoting the uncontrolled proliferation of atypical keratinocytes. However, similar mutations can also be found in normal keratinocytes, mainly in chronically sun-exposed skin [2]. Therefore, other factors—including epigenetic modifications, viral infections, or microenvironmental changes—can favor cSCC development and progression [7]. Deeper studies concerning the characterization of cSCC pathogenic mechanisms have been useful to identify new drug target and develop effective therapies.

### 8.1. Genetic Mutations

cSCC is one of the cancers with the highest mutation rate. The commonest mutated genes belong to pathways involved in cell cycle regulation, apoptosis, senescence, differentiation, and mitogenic/survival signaling [63].

#### 8.1.1. Cell Cycle Regulation, Apoptosis, and Senescence

##### TP53

The most frequently altered gene in cSCC is *TP53*, which encodes the tumor-suppressor protein p53. p53 is a transcription factor that plays an important role in maintaining genomic stability. Following several cellular stresses, p53 regulates the expression of its target genes and therefore induces cell cycle arrest, apoptosis, senescence, DNA repair, or changes in metabolism [64,65].

The inactivation of p53 in cSCC is mainly due to gene mutations or interactions with viral proteins such as HPV E6 [66] (Figure 6a). Missense “hot spots” mutations are mainly characterized by UV-signature (i.e., C > T or CC > TT transitions), which enable keratinocytes to prevent apoptosis and to promote clonal expansion of *TP53* mutated cells [66]. The role of p53 in UV-B-induced carcinogenesis has been confirmed in animal models [67].

The mutation in *TP53* sequence is an early event in cSCC pathogenesis, occurs in 54–95% of cases, and is responsible for the great genomic instability of these tumors [68,69,70]. Indeed, mutations are reported both in early lesions, such as AKs and in situ SCCs (7–48% of samples), and metastatic cSCCs (79%) [69,70,71]. Notably, normal human skin, especially sun-exposed areas in aging individuals, contains clusters of epidermal cells with *TP53* mutations that can increase in size over time [70]. Higher mutation frequency has been found in metastatic tumors compared to primary lesions (85% vs. 54%; *p* < 0.002), consistent with p53 function also against cancer progression [72].

##### CDKN2A Locus Gene and pRb Pathway

The *CDKN2A* locus gene encodes two alternatively spliced proteins, p16INK4a and p14ARF, which inhibit cell cycle progression and proliferation through the retinoblastoma (pRb) and p53 pathways, respectively (Figure 6A). Following mitogenic stimuli, cyclin D1 promotes the G1- to S-phase transition, activating cyclin-dependent kinase 4 (CDK4) or CDK6. These kinases phosphorylate pRb, thus inducing pRb-E2Fs dissociation and transcription of E2F-target genes. p16INK4a is considered a tumor suppressor gene since it directly binds CDKs, inhibiting their kinase activity, preventing pRb phosphorylation and E2F-mediated transcription, and therefore inducing cell-cycle block and senescence [73]. p14ARF is a tumor suppressor induced in response to elevated mitogenic stimulation and able to inhibit HDM2 forming a stable complex. HDM2 functions are binding p53 and promoting its degradation. Therefore, p14ARF antagonizes with HDM2, permitting the transcriptional activity of p53 that leads to cell cycle arrest or apoptosis [74]. 

The inactivation of the *CDKN2A* locus can be due to loss of heterozygosity, point mutations, and promoter hypermethylation and leads to unrestrained cell cycling and uncontrolled cell growth [75]. Loss of heterozygosity or point mutations has been found in 21–62% of cSCCs, whereas promoter hypermethylation has been identified in 35–78% of cases [75,76]. *CDKN2A* mutations were observed in 31% of the metastases and their primary tumors [77].

The pRb is a tumor suppressor gene that stabilizes constitutive heterochromatin beyond E2F transcription regulation. However, few studies report *RB1* gene inactivation. Loss of protein expression was reported in 8% of AKs and 16% of cSCC cases [78].

Cyclin D1 overexpression seems an early event in cSCC carcinogenesis since it has been identified also in AKs. Specifically, cyclin D1 is overexpressed in 43–46% of BD and AKs and 60–71% of cSCC cases [78,79]. A positive correlation between cyclin D1 overexpression and invasion depth and metastasis has been reported [80]. However, there is no correlation between cyclin D1 overexpression and the degree of cSCC differentiation [79,81,82]. Overall, larger studies are needed to confirm p16 or cyclin D1 as cSCC prognostic biomarkers.

##### hTERT Promoter

Progressive shortening of telomeres occurs during each cell division. Telomerase is a ribonucleoprotein complex that synthesizes telomeric DNA (TTAGGG hexamers) required to maintain telomere length [83]. The activation of the hTERT, which is the catalytic subunit of telomerase, can occur through promoter (*TERTp*) mutations that create de novo binding sites for the ETS transcription factors family. Therefore, the activation of hTERT increases telomere length and prevents senescence or apoptosis of mutated cells [84,85].

*TERTp* mutations with UV signature have been described in cSCC [65]. *TERTp* mutations are more frequent in cSCC (50%) than in Bowen’s disease (20%), suggesting an important role in tumor progression [85]. Notably, *TERTp* mutations have been found in 31.6% of cSCCs, mostly related to recurrent and metastatic lesions [65]. However, further studies are necessary to confirm *TERTp* mutations’ value as prognostic markers for cSCC.

#### 8.1.2. Keratinocyte Differentiation 

##### NOTCH1 and NOTCH2

The Notch family includes single-pass transmembrane receptors composed by an extracellular ligand-binding domain with multiple EGF-like repeats and an intracellular domain that mediates the transcription of target genes. The *NOTCH1* gene is a direct target of p53 and is involved in cell-cycle exit and keratinocyte differentiation [86] (Figure 6B). The Notch signaling sustains the expression of transcription factor interferon regulatory factor 6 (IRF6) that contributes to the transcription of growth/differentiation-related genes. IRF6 is also a p63 transcriptional target that restrains keratinocyte proliferation by inducing proteasome-mediated degradation of the isoform ΔNp63 in a feedback regulatory loop [87]. Indeed, IRF6 exhibits tumor suppressor activity in mouse models and is downregulated in human SCCs [88]. This crosstalk between the p53 pathway and Notch and p63 pathways connects external damaging signals with the control of stemness/differentiation balancing and therefore is deregulated during SCC development.

The most common alterations of Notch signaling are inactivating mutations, copy number aberrations, and loss of heterozygosity that result in loss of function. Thus, Notch is considered a family of tumor suppressors. NOTCH1 loss of function can also upregulate the Wnt/beta-catenin pathway, promoting tumor development [89]. *NOTCH1* inactivating mutations have been identified in 42–75% of aggressive or advanced stage cSCCs, whereas *NOTCH2* mutations have been observed in 18–51% of cases [71,90,91]. Notably, a recent exome sequencing study reported *NOTCH* mutations in 82% of cases in cSCC and 70% in normal skin, indicating *NOTCH1* and *NOTCH2* mutation as an early event in cSCC pathogenesis [90]. The precocious role of NOTCH1 in carcinogenesis has been confirmed in animal models. Notch signaling inhibition has been associated with impaired differentiation, development, and progression of cSCC [89]. FBXW7 is part of the ubiquitin ligase complex that mediates NOTCH1 degradation. *FBXW7* alterations were found in 7% of SCC cases [91]. Thus, *FBXW7* mutations can be an alternative mechanism for Notch inactivation. 

##### TP63

The p63 protein, together with its homologue p73, is part of the p53 family of transcription factors; these proteins are characterized by a conserved gene structure and a high sequence homology. *TP63* encodes for different protein isoforms depending on the use of alternative promoters (i.e., TAp63 and ΔNp63) or on alternative splicing occurring at the C-terminus of the mRNA, giving rise to at least six isoforms (i.e., TAp63α, TAp63β, TAp63γ, ΔNp63α, ΔNp63β, and ΔNp63γ) [92,93,94].

The ΔNp63 isoform is the most expressed in tissues of ectodermal origin, such as epidermis, skin appendages, simple epithelia, and thymus. In normal epidermis, ΔNp63 is expressed in the basal layer where it represses the expression of *CDKN1A* (also known as p21) and *HES1* (a member of Notch pathway), thus supporting keratinocytes proliferation [95] (Figure 6B). 

An altered expression of p63 appears as a common feature in squamous cancers such as cSCC. p63 is involved in SCC development through disruption of numerous molecular networks involved in the regulation of proliferation, differentiation senescence, cell adhesion, migration, and invasion. For instance, p63 acts as an oncogene enhancing Ras-driven tumorigenesis of undifferentiated cSCC [96]. Moreover, ΔNp63 also delays keratinocytes senescence through direct binding to the p16INK4A and p19ARF promoters to repress their expression in cSCC development [97]. 

The protein p63 is highly expressed in cSCC (70–100% of cases), including AKs. Different from *TP53*, the *TP63* gene is rarely mutated in human cancers. However, several genetic and molecular alterations (i.e., p53, NOTCH1, IRF6 dysregulation) are able to increase p63 expression or enhance its transcriptional activity [98] (Figure 6B). p63 exhibits higher expression on the periphery of the tumor, gradually reducing toward the well differentiated center. Thus, p63 expression can be a strong predictor of SCC poor differentiation. *TP63* was amplified in 24% of metastatic cSCC cases [65]. However, there are no specific studies addressing the putative prognostic value of *TP63* in cSCC.

##### RIPK4

RIPK4 is a serine/threonine protein kinase that interacts with protein kinase C-delta involved in keratinocyte differentiation [99]. RIPK4 promotes differentiation, at least in part by phosphorylation of IRF6 (Figure 6B). The RIPK4-IRF6 signaling axis may also regulate the inflammation through the expression of specific proinflammatory cytokines (e.g., Chemokine (C-C motif) ligand 5 (CCL5) and **C**-X-C motif chemokine 11 (CXCL11)). 

*RIPK4* mutations have been found in several SCCs. Recurrent *RIPK4* mutations were identified in 24% of metastatic cSCC [71]. However, additional studies are needed to investigate the relevance of *RIPK4* as prognostic marker. 

#### 8.1.3. Mitogenic/Survival Signaling Pathways 

##### Epidermal Growth Factor Receptor

The EGFR gene encodes a transmembrane glycoprotein of the ErbB family of tyrosine kinase receptors (RTKs). The ligand-RTK binding promotes homo- or heterodimerization of the receptor, autophosphorylation of tyrosine, and activation of downstream ERK and PI3K/AKT pathways that control cell growth, proliferation, inhibition of apoptosis, and angiogenesis [65] (Figure 6C). 

EGFR overexpression, which is consistent with constitutive pathway activation, was observed in 43–73% of cSCC cases and associated with more aggressive phenotype and poor prognosis. However, *EGFR* activating mutations were identified with low frequency in cSCC (2.5 to 5%) [100,101,102]. Moreover, *EGFR* gene amplification was found only in few cSCC, and EGFR protein overexpression did not correlate with transcript levels [102]. Therefore, the overexpression of EGFR may depend on reduced degradation and dephosphorylation [103]. Notably, EGFR activation downregulates the expression of p53 and, in turn, NOTCH1 [65]. The RTKs, including EGFR and downstream pathways, can both be targeted with several drugs (cetuximab, panitumumab, erlotinib, gefitinib, and dasatinib) to inhibit cSCC progression [63].

##### RAS-RAF-MEK-ERK Pathway

The RAS-RAF-MEK-ERK cascade is the mitogen-activated protein kinase (MAPK) signaling pathway. The ligand-RTK binding is followed by GTPase protein RAS activation. The active RAS promotes the formation of RAF dimers that activate MEK-ERK cascade through phosphorylation. Several feedback loops regulate this kinase cascade [65]. ERK act on multiple cytosolic and nuclear targets, including kinases, cytoskeletal protein, and transcription factors [66] (Figure 6C). For instance, ERK regulates the transcriptional activity of ETS1 that, in turn, promotes transcription of several key players of tumor development and progression, including genes involved in cellular proliferation and survival (bcl-2, caspase-1, p16INK4a, p21, p53) and extracellular matrix remodeling and angiogenesis (metalloproteases) [30].

Although the three members of the RAS family (*HRAS*, *KRAS* and *NRAS*) are frequently mutated in human tumors, *RAS* mutations occur with low frequency in cSCCs. Nevertheless, mutations were mainly found in the *HRAS* gene [66]. *HRAS*-activating mutations were identified in 3–20% of cSCCs [71,76,99]. *HRAS* deletions were found in 10% of primary cSCCs and in 10% of metastatic cSCCs [71,76]. Notably, *HRAS* mutations were detected at higher frequency in cSCC lesions arising in melanoma patients treated with BRAF-inhibitors. This may be due to paradoxical MAPK activation in keratinocytes with preexisting *HRAS* mutations [104,105]. The increased RAS activation frequently observed in cSCC may also result from upstream RTK stimulation or RASA1 inactivation. RASA1 is a negative regulator of the pro-oncogene RAS and mutations in its gene were identified in 13% of cSCCs [65]. *B-RAF* gene mutations are rare events in cSCC [65]. ETS-1 was overexpressed in poorly differentiated and metastatic SCC compared to in situ and well differentiated SCC [106,107]. 

##### PI3K/AKT/mTOR Pathway

The PI3K/AKT/mTOR pathway is the survival signaling pathway downstream to RTKs and RAS (Figure 6C). Once activated, the lipid kinase PI3K converts plasma membrane PIP2 into PIP3, leading to the activation of the serine/threonine kinase AKT that, in turn, activates mTOR [65]. The mTOR acts as a sensor of nutrients and regulates cell growth through induction of RNA translation and protein synthesis [108]. This signaling cascade is negatively regulated by Phosphatase and tensin homolog (PTEN) through dephosphorylation of PIP3 to PIP2 [109].

In mouse models, activating mutations or amplification of *PIK3CA* gene, which encodes the catalytic subunit of PI3K, or *PTEN* loss of function can activate of the PI3K/AKT/mTOR pathway, promoting the development and progression of cSCC [65,110].

*PIK3CA* mutations were identified in 10% of cSCCs. Activating mutations in the genes of the RAS/RTK/PI3K pathway were identified in 45% of metastatic cSCCs and significantly correlated with worse progression-free survival [71]. *PTEN* gene alterations, including inactivating mutations and homozygous loss of function, have been found in 7–25% of cSCCs [71,76].

### 8.2. Epigenetic Modifications

Environmental factors can alter the epigenetic status of the cells. Epigenetic consists of molecular mechanisms that regulate gene expression without modification of the DNA sequence. Epigenetic alterations include DNA methylation and histone modifications (i.e., methylation, acetylation, phosphorylation, ubiquitination, and chromatin remodeling) [111].

DNA methylation is established and maintained by DNA methyltransferases and results in a covalent modification of cytosines within CpG dinucleotides. Changes in genomic DNA methylation associated with cancer include gene-specific hyper- or hypomethylation and global DNA methylation modifications. For instance, UV irradiation induces large hypomethylated blocks in healthy chronic sun-exposed epidermis [112]. The methylome deregulation is a hallmark of the development and progression of human cancers, and methylation signatures can be used as biomarkers for tumor diagnosis and prognosis [111]. 

An association of cSCCs and gene-specific promoter hypermethylation has been reported. Specifically, they are genes involved, in cell cycle (*CDKN2A*), positive regulation of apoptosis (*ASC, G0S2* and *DAPK1*), Wnt signalling (*SFRPs* and *FRZB*), transcription factor forkhead box *(FOX)E*, and adhesion molecules (cadherin *CDH1* and *CDH13*) [113,114,115,116]. Promoter methylation of p16(INK4a) and p14(ARF) was detected in 36% and 42% in cSCC, respectively. The absence of protein expression was confirmed by immunohistochemistry in 82% of the lesions with biallelic inactivating events [75]. E-cadherin (*CDH1*), a key protein of adherent junction complex, is downregulated in epithelial–mesenchymal transition. E-cadherin promoter hypermethylation was found in cSCC (85%), in situ cSCC (50%), AK (44%), and normal skin (22%). Downregulation of E-cadherin is associated with increased tumor invasiveness, metastatic potential, and advanced stage of cSCC [116,117].

Global methylation patterns corresponding to cancer methylomes were identified in AKs and cSCCs when compared to normal skin. However, no differences between AK and cSCC were observed [118]. Recently, a global-scale approach to investigate the DNA-methylation profile in patients at different stages (i.e., AK, low-risk invasive, high-risk nonmetastatic, and metastatic cSCC) identified a minimal methylation signature to discriminate different stages, especially distinguishing low-risk vs. high-risk stages. Moreover, a prognostic prediction model in cSCC patients identified a methylation signature able to predict the overall survival of patients [119].

Multiple types of histone modifications are catalyzed by several enzyme families. The most well characterized modifications are acetylation and methylation of histones H3 and H4 that directly alter chromatin condensation and gene transcription.

Acetylation is catalyzed by histone acetyltransferases (HATs) that add an acetyl group to lysine amino acids on the histone tail, loosening chromatin to promote gene activation. On the contrary, histone deacetylases (HDACs) remove the acetyl group, allowing chromatin condensation and, in turn, gene inactivation. The HAT p300 upregulation plays an important role in the development and progression of cSCC. High p300 expression correlates with aggressive features of cSCC, suggesting that p300 may be a promising biomarker to predict clinical outcomes [73]. Interestingly, ΔNp63α controls the epigenetic landscape of keratinocytes recruiting epigenetic modulators and chromatin remodelers to regulate the transcription of several genes involved in proliferation, differentiation, and adhesion. During SCC development, ΔNp63α activates genes transcription through physically interacting with several epigenetic modulators (e.g., p300) to enhance chromatin accessibility and activate oncogenic transcripts linked to poor prognosis. Moreover, ΔNp63α interacts with repressive epigenetic regulators to represses gene transcription during tumorigenesis of various SCCs. For instance, ΔNp63α/HDAC complex acts as a direct repressor of the apoptotic transcriptional program in various SCCs [98].

Methylation occurs on lysine and arginine amino acids by histone methyltransferases, and modification effect depends upon which residue is modified. For instance, trimethylation of histone H3 on lysine 4 (H3K4me3) activates transcription, whereas trimethylation of lysine 27 or lysine 9 on histone H3 represses transcription [120]. Polycomb group proteins (PcG) are an important family of histone modifiers extensively studied in skin cancer. The PcG enzyme EZH2 is the major histone methyltransferase and controls proliferative potential of self-renewing keratinocytes by repressing the CDK2A locus. EZH2 is frequently mutated in cancer, and its overexpression is associated with malignant cSCC progression. EZH2 repress the innate inflammatory cSCC function and impair tumor immunosurveillance by modulation of NF-kB signaling [121,122,123]. The PcG subunit BMI-1 is also highly expressed in cSCCs [73].

The type 2 lysine methyltransferases KMT2C and KMT2D form the core of KMT2C/D COMPASS complexes involved in transcriptional regulation through the targeted modification of histone H3. *KMT2C* and *KMT2D* genes show high rates of mutations in cSCC. Mutations in *KMT2C* and *KMT2D* have been found in both primary cSCC (36% and 31%, respectively) and metastatic samples (43% and 62%, respectively) [72]. Moreover, aggressive cSCCs display frequent inactivating mutations in *KMT2C* (38.5%) and *KMT2D* (69.2%) genes [99].

### 8.3. Viral Pathogenesis

HPV is a double-stranded DNA virus that infects squamous epithelia. HPVs have been classified in 5 genera (alpha, beta, gamma, mu, and nu). High-risk alpha mucosal HPVs (HPV-16 and HPV-18) are well established causative agents for cervical and oropharyngeal cancers. Mechanism of HPV oncogenesis relies on viral proteins E6 and E7. E6 and E7 bind p53 and RB1, respectively, and promote their proteasomal degradation, leading to a loss of tumor suppressor genes and uncontrolled proliferation [124,125].

The presence of beta-HPV DNA in cSCC samples and the detection of antibodies against HPV in patients with cSCC indicate that beta-HPV types can be involved in cSCC pathogenesis [124,126]. HPV-5 and HPV-8 have been reported in 90% of cSCC cases as a rare genetic disease termed “epidermodysplasia verruciformis” (EV) [124]. Beta-HPV were also identified in cSCC arising in chronically immunosuppressed patients. Beta-HPV infection seems to play an important role in early steps of carcinogenesis, but not in tumor progression. This hypothesis is supported by the identification of an increased viral load in AKs compared to cSCCs [127,128]. Moreover, beta-HPV DNA was identified in 9% of primary cSCC tumors and in 13% of metastases, indicating that HPV does not play a relevant role in advanced stages of cSCCs [129]. Tumor growth also occurs in the absence of the viral genome [124]. The high incidence of beta-HPV types in healthy skin indicates that some cofactors contribute to its pathogenesis. Beta-HPV are components of the normal flora, but, under the influence of certain cofactors, the virus may trigger tumor development [20]. Dysregulation of the immune system (chronic inflammation and immunosuppression), environmental factors (UV radiation), and genetic factors are the most important cofactors. Notably, concomitant seropositivity for high-risk mucosal HPV-16 and cutaneous HPV types increases the risk of recurrent squamous cell carcinoma of the skin [21]. Therefore, HPV can play a role at early stages of cSCC development by potentiating the deleterious effects of other cofactors [124].

### 8.4. Tumor Microenvironment

Tumor microenvironment (TME) of solid tumors is a complex system of molecules, including mediators of senescence-associated secretory phenotype, or SASP (i.e., cytokines, growth factors, and metalloproteinases), and heterogeneous population of cells comprising tumor cells and nearby stromal cells such as fibroblasts, endothelial cells, and several inflammatory and immune cells. Cells residing in the TME gain specific pro-tumorigenic phenotypes and functions. In aged tissue, the synergic contribution of genomic instability, SASP, and decline of the immune system function induces accumulation of senescent cells, contributing to worsening the senescence response efficacy in tumor suppression and inducing a chronic inflammatory status [7,130,131].

The TME is important in the carcinogenesis of cSCC [7,132]. cSCC emerges from the interplay of nascent neoplastic keratinocytes with other stromal cell types hosted in the local microenvironment. Inflammation acts as tumor promoter and modifies these cell–cell interactions by epigenetic reprogramming, damaging DNA, promoting hypoxia and angiogenesis, activating cancer-associated fibroblasts (CAFs), recruiting regulatory immune cells, and inhibiting antitumor immune surveillance. This dynamic network system stimulates tumor growth, migration, and invasion [7,132,133].

In organotypic cultures, the epidermal compartment generated by aged keratinocytes displays cellular hyperplasia and may invade the matrix only if it contains CAFs that recreate a permissive microenvironment. On the contrary, the normal microenvironment has tumor-suppressive properties that counteract cSCC invasion [134].

In SCC, p63 cooperates with NF-κB members c-Rel and rel-A to transcriptionally regulate a cohort of proinflammatory cytokines—such us IL-1, IL6, IL8—able to attract inflammatory infiltrating cells, such us neutrophils and macrophages [98]. This tumor microenvironment is responsible of the malignant behavior and poor prognosis of this type of cancer [135]. Coherently with the recruitment of inflammatory cells via proinflammatory cytokines release, p63 can sustain tumor associated angiogenesis. In osteosarcoma cells, the ectopic expression of p63 enhances vascular endothelial growth factor (VEGF) expression through proinflammatory cytokines expression. However, in SCC, this pathway has not been proved and investigated. Different studies suggest an association between p63 and lymphangiogenesis in SCC, and, in fact, p63 can enhance the expression of human beta defensins (HbD1, HbD2 and HbD4). These antimicrobial factors can stimulate the migration of lymphatic endothelial cells and determine the increase of the number of lymphatic vessels in epithelial tumors [136]. These findings demonstrate that p63 contribution to SCC development is at a global level, also controlling the tumor microenvironment.

## 9. Treatment of cSCC

### 9.1. Local Treatment

The primary goal of treatment of cSCC is the complete removal of the tumor with the maximal functional and cosmetic preservation. Most cSCCs are successfully treated with surgical excision alone, with a good prognosis and cure rates greater than 90% [6]. Although the surgical approach is the most effective and efficient treatment, traditional techniques (i.e., curettage, electrodessication, cryosurgery, lasers, photodynamic therapy (PDT)) are available for noninvasive cSCC such as BD. Also, RT represents a good alternative and curative treatment strategy compared to surgery for small cSCCs.

#### 9.1.1. Surgery for cSCC

Conventional surgery with safety margins and micrographically controlled surgery (MCS) are the two different surgical approaches that may be utilized in patients with primary cSCC. MCS gives the highest rate of R0 resection (i.e., no cancer cells seen microscopically at the primary tumor site), above 90%, with lower recurrence rates (0–4%) compared to conventional surgery (3.1–8.0%) [137,138,139]. Regarding MCS, two different techniques have been described: the Mohs micrographic surgery (MMS), based on intraoperative frozen sections; and procedures based on paraffin-embedded section analysis (i.e., “slow” Mohs, 3D histology, and complete peripheral and deep margin assessment). These approaches are generally reserved for patients with high-risk tumors in order to obtain a complete tumor resection with optimal anatomic and functional preservation.

#### 9.1.2. Standard Excision with Postoperative Margin Assessment

A common therapeutic approach for cSCC is the standard excision followed by postoperative margin assessment. Safety margins with clinical normal-appearing tissue around the tumor and negative margins as reported by the pathology are needed to minimize the risk of local recurrence and metastasis [140,141,142]. It has been reported that this technique guarantees a five-year disease-free rate of 91% or higher for cSCC [143,144].

Safety excision margins should be decided according to the risk of subclinical extensions and recurrence of the tumor depending on the low- or high-risk factors of the cSCC [145]. For low-risk cSCCs with a diameter of less than 2 cm, a margin of 4 mm has achieved cure rates of 95–97% in prospective studies [141]. Guidelines indicate margins between 4 mm and 6 mm for tumors lacking high-risk features [8,146,147,148]. The European consensus group proposed a 5 mm margin for low-risk cSCC [8]. Otherwise, for high-risk cSCC, although broader safety margins are needed, no unified recommendation on appropriate safety margins are available. The American Academy of Dermatology (AAD) and National Comprehensive Cancer Network (NCCN) guidelines recommend safety margins for cSCC in high-risk locations (scalp, ears, eyelids, nose, lips) or with other high-risk features (histologic grade ≥ 2, invasion of subcutaneous tissue), with a diameter <1 cm, 1 to 1.9 cm, and ≥2 cm margins of at least 4 mm, 6 mm, and 9 mm, respectively [3,42,149]. For cSCCs >2 cm in maximum clinical diameter and/or with other high-risk factors, an excision margin of at least 5 mm is needed [14,141,150]. The European consensus group proposes 6–10 mm safety margins for high-risk cSCCs [141]. Furthermore, in patients affected by multiple invasive cSCCs (i.e., on the dorsal hand or scalp), en bloc excision of the involved field with subsequent skin grafting could be an effective surgical strategy, with a depth of excision that should include the subcutaneous tissue [8]. A re-excision should be done for operable cases in the event of positive margins, while wider excision should be considered when margins appear more limited than the recommended safety margins due to the tissue shrinkage [8].

#### 9.1.3. Micrographically Controlled Surgery

MCS is a surgical technique of excision of the skin tumor, processing skin tissue in horizontal sections and examining them under a microscope until all borders are tumor-free. Two techniques are available in Europe: MMS, and 3D histology [8,151]. The two techniques differ because the first utilizes frozen sections while the second uses paraffin sections [151]. MMS is more time-consuming, labor-intensive, and expensive, compared to conventional excision. A retrospective study including 579 patients with 672 cSCCs of the head and neck (380 treated with MMS and 292 with standard excision) reported that MMS could be superior to standard excision for cSCC of the head and neck because of the lower rate of recurrence [152].

#### 9.1.4. Surgery for Regional Nodal Disease

Guidelines regarding the management of regional nodal disease in patients with cSCC are scanty and based on studies made in head and neck mucosal cSCC [153]. Patients with nodal metastases due to cSCC should be treated surgically, similar to patients affected by other skin tumors such as melanoma or Merkel cell carcinoma. When surgery is not indicated, e.g., for patient-related factors, a nonsurgical approach by a multidisciplinary group should be considered.

The indicated surgical treatment in case of nodal metastasis is therapeutic regional lymph node dissection [3,8,150]. Elective or prophylactic lymph node dissection in cases of patients affected by cSCC with negative lymph node is not recommended, given the low rate of nodal metastases, the high morbidity, and the limited evidence in patients with mucosal head and neck cSCC [154,155].

#### 9.1.5. Treatment Alternatives to Surgery for Limited Cases of Low-Risk cSCC

##### Curettage and Electrodessication

NCCN guidelines reported that curettage and electrodessication may be considered for small and low-risk primary cSCC [3,42]. Cases of cSCC localized on terminal hair-bearing skin—such as the scalp, pubic, or axillary regions, and on the beard area in men— must be excluded from this treatment [3,4,42].

##### Cryotherapy and PDT

NCCN guidelines reported that cryotherapy could be an option in selected cases of low-risk cSCC [3,42]. Otherwise, there is scarce evidence regarding the efficacy of PDT for invasive cSCC, and it should not be recommended in these cases [156].

##### Intralesional Cytostatic Drugs

Intralesional cytostatic drugs such as methotrexate, 5-fluorouracil, bleomycin, or interferon could be considered in case of clear KA. In case of incomplete regression of the tumor, the lesion should be completely excised because it could hide an invasive cSCC [8].

### 9.2. Systemic Treatment, RT and Electrochemotherapy

#### 9.2.1. Systemic Treatment

The cSCC is characterized by a high mutational burden. Moreover, the tumor immune microenvironment is characterized by changes in immune cell populations, immune checkpoint expression, and altered balance of the immune milieu in favor of immunosuppression, allowing tumor escape from immune surveillance [2,157]. Additionally, the increased risk of cSCC among people with immunodepressing conditions reflects an important role for immune surveillance in the pathogenesis of this cancer [158]. Thus, cSCC displays the hallmarks of solid tumors responsive to systemic immunotherapy [2,157]. Furthermore, the expression of the programmed death (PD)-ligand 1 (PD-L1) expression or the presence of the interferon (IFN)-γ gene signature are potential biomarkers in predicting response to immune checkpoint inhibitors (ICI) [157].

An ICI, cemiplimab, is available as systemic therapy for the treatment of metastatic (m)cSCC and for the locally advanced (la)cSCC not amenable to surgical or RT treatment. It is the first approved treatment in the USA and Europe for these patients [159].

Cemiplimab is a high-affinity, human monoclonal antibody directed against anti-programmed death 1 (PD-1). In the phase 1 study, the response rate in patients with mcSSC or lacSSC was 50%, while in a phase 2 study the response rate was 47% in a metastatic disease cohort. In subgroup analyses, a similar efficacy was observed in patients with regional metastatic disease and in those with distant metastatic disease [158]. In a further recent single-arm phase 2 trial, 34 of 78 patients with lacSCC, for whom there was no widely accepted standard of care, showed an objective response to cemiplimab (44%), which had an acceptable safety profile. Specifically, 10 patients (13%) displayed a complete response and 24 (31%) a partial response. [160]. The most common adverse events were hypertension, diarrhea, fatigue, nausea, constipation, and rash [158].

The approved fixed dose regimen is 350 mg intravenously every three weeks. Durable responses and similar safety profile have been observed in both weight-based and fixed-dosing groups [161].

Patients not eligible for anti-PD-1 treatment or patients refractory to anti-PD-1 may be offered EGFR inhibitors and/or chemotherapies [162].

EGFR is highly expressed in cSCC, but its role as a target of therapeutic agents is still unclear. Two monoclonal antibodies to EGFR, cetuximab and panitumumab, were evaluated in clinical trials. In particular, cetuximab was tested in different settings: as first line in patients with metastatic disease, and in combination with RT in patients with lacSCC who were not candidates for a surgical approach due to comorbidity, or who were not candidates due to the impossibility of radical intervention, or with patients at risk of aesthetic/functional damage. Additionally, as an adjuvant treatment, it has been tested as a single agent or in combination with platinum salts and 5-fluoruracil [162].

Cetuximab monotherapy was evaluated in a retrospective study on 58 patients, showing good safety even in elderly subjects. The objective response rate (ORR) was 53% and 42%, respectively, at 6 and 12 weeks. The median progression-free survival (PFS) and overall survival were 9.7 months and 17.5 months, respectively [163].

Panitumumab also showed an advantage in the treatment of lacSCC. The studies have shown results in terms of response, but they still need to be validated on a larger and more homogeneous series [162].

In a phase II study, an oral anti-EGFR agent, gefitinib, was evaluated in 40 patients with advanced cSCC at a dose of 25 mg daily. The overall response rate was 16% and the median PFS was 3.8 months [164].

Chemotherapic agents used in cSCC treatment are cisplatin, doxorubicin, 5-fluorouracil (5-FU), methotrexate, and bleomycin. Their role as systemic treatments is limited predominantly to some specific situations in advanced cSCC treatment, particularly in the absence of valid therapeutic alternatives, since there is not strong enough evidence of outcome improvement for their use (alone or in combination with retinoids or interferon-alpha), neither as neoadjuvant nor as adjuvant therapy [165,166].

In advanced cSCC, evidence on chemotherapeutic treatment is limited to several case-series and some phase 2 studies [167]. Systemic therapy impact on 119 patients with lacSCC observed in 28 studies was analyzed in Behshad et al. [168]. A response of 50% was observed for platinum-based monotherapy, with higher rates when combined with 5-FU. The best results were, however, obtained in combination with RT. Toxicity (i.e., nausea and diarrhea) was not negligible, especially for polytherapy [168].

A systemic review by Trodello et al. analyzed 60 cases of mcSCC reported in the literature from 1989 to 2014 [169]. Patients had obtained a complete response of 22%, an overall response of 45%, and a median disease-free survival (DFS) of 14.6 months (range 3–112 months) [170].

Since first-line chemotherapy provides lower response and greater toxicity compared to immune checkpoint inhibitors treatment of advanced cSCC, it should be limited for use in patients not eligible for immune therapy treatment (i.e., patients with recent organ transplantation or other immunosuppressive conditions). In case of tumor progression after or during immunotherapy, chemotherapy can be considered as a second-line treatment [167].

#### 9.2.2. Radical RT

RT may be used in cSCC as an alternative primary treatment for patients who are not eligible for surgery because of cSCC features (i.e., locally advanced and/or unresectable lesion), or because of patient peculiarity (e.g., multimorbidity, frail elderly patient, high-risk surgical patient), or because the patient elected to refuse surgery. Primary RT may also be considered for patients with cSCC in an esthetically (i.e., neck or head) or functionally (i.e., lips or eyelids) sensitive location of the tumor. Definitive radical primary RT can be also a good alternative therapeutic choice for small cSCC [171,172].

Since RT has lower rates of cure and presents a considerable number of cases with aggressive posttreatment recurrence, surgery should be preferred wherever possible. In fact, an average rate of local tumor recurrence after the first-line RT in 1018 primary cSCC patients was approximately 6.4% according to a meta-analysis of 14 observational studies [173]. However, there is no strong scientific evidence comparing patient survival after surgery and after definitive RT [8,172].

RT adverse effects, though relatively rare, include radiodermatitis, hypo/hyperpigmentation, and telangiectasia. Furthermore, patients with preexistent genetic disorders present a higher risk of radiosensitivity (i.e., Gorlin syndrome or ataxia telangiectasia), and therefore primary RT is not recommended in these cases [174]. Young age may also be a conditional contraindication for primary RT because late chronic adverse effects such as telangiectasia become more visible with the passing of time [175].

#### 9.2.3. Adjuvant Therapy

Adjuvant therapy is an additional treatment to reduce the risk of recurrence after the complete surgical excision of high risk cSCC (Table 1) [176].

This therapy can be systemic or RT. Actually, there are no drugs approved for the treatment of cSCC as adjuvant therapy. The most common chemotherapy agents used were 5-fluoruracil, platinum salts, or paclitaxel, with scarce results [1,177].

In the prospective trial of adjuvant cetuximab and radiation for lacSCC of the head and neck, overall survival at 2 years was 75% and was 67.5% at 5 years. Two-year DFS was 70.8%, and five-year DFS was 55.9%. Cetuximab was well tolerated and its use plus RT may be a reasonable approach in cSCC of the head and neck for patients who are at high risk of disease recurrence or progression, but further studies are needed [178].

A recent phase 2 study indicated that cemiplimab neoadjuvant therapy is well tolerated and induced histological responses in 70% of the patients. Ongoing trials are evaluating the potential contribution of ICIs as adjuvant therapy for high-risk cSCCs [179].

Postoperative RT (PORT) should be considered in cSCC patients with high-risk tumor recurrence factors, including positive post-excision margins that cannot be re-excised and in cases of perineural invasion [174].

The scientific evidence regarding the impact of treating positive margins postoperative cSCC with RT is rather poor, although incomplete excision is proved to be a high-risk factor of recurrence and although a conspicuous rate of excisions (7–18%) presents positive surgical margins [174]. However, a study of PORT in the postoperative positive margins cSCCs of the lower lip presents a 6% rate of local recurrence compared to a 64% rate of recurrence for nontreated cSCCs [173].

Use of PORT incidental findings of perineural invasion is highly recommended by various guidelines, although the scientific evidence is controversial, probably because of lack of specifically designed studies: most studies are in fact retrospective, and often there is no distinction between cases with negative or positive postoperative margins, which could significantly affect outcomes [180].

PORT is highly recommended for cSSC patients with lymph node and/or parotid metastasis (except in cases of just one small lymph node involvement without extracapsular spread) [174]. A 2005 study that evaluated 167 cSSC patients with lymph node involvement had shown significantly lower recurrence rate (20% vs. 43%) and higher five-year DFS rate (73% vs. 54%) in patients who underwent PORT, compared to patients with surgery only (14520114). More recent studies, e.g., a retrospective analysis by Harris et al. of OS and DFS in 349 patients with head and neck cSCC with perineural invasion or regional metastasis, confirmed that PORT may improve life quality and reduce the risk of death for this setting [181].

PORT can also be considered for other high-risk tumor factors, such as a lesion with thickness >2 mm (and especially 6 mm or more), or a lesion with a diameter >2 cm or poor differentiation, and for immunodepressed patients [180].

#### 9.2.4. Neoadjuvant Therapy

The efficacy of cemiplimab in the neoadjuvant treatment of cSCC of the head and neck was evaluated. Anti-PD-1 was administered every 3 weeks for two cycles. On 20 enrolled patients, pathologic complete response was obtained in 11 (55%) patients and major pathology response in 3 (15%) patients [182].

However, there is currently no treatment indicated in the neoadjuvant setting of the cSCC.

#### 9.2.5. Electrochemotherapy

Electrochemotherapy is one of the alternative treatments of unresectable cSCCs and can be used for disease control and/or reduction of tumor progression. An intravenous injection of a chemotherapy agent (i.e., cisplatin or bleomycin) is combined with local electric pulses that permeabilize tumor cell membranes, with a dramatic increase of its cytotoxicity [183].

According to some retrospective studies and one meta-analysis, 20–70% of patients treated with electrochemotherapy showed good local response and disease control [184,185,186]. In the multi-institutional prospective study European Research on Electrochemotherapy in Head and Neck Cancer (EURECA) of the effectiveness of electrochemotherapy that considered 47 patients affected by cSCC, the rate of complete response at 2-months follow-up was 55% (26 patients), with only a 4% rate of progression. Toxicity was predominantly local (i.e., hyperpigmentation and skin ulceration) [170].

## 10. Prevention of cSCC

An early diagnosis and a correct treatment are pivotal to improve cSCC outcomes, but prevention plays a crucial role in reducing the incidence and improving the prognosis of cSCC. The most important prevention measure is represented by the reduction of UV exposure, both professional and recreational [187,188]. Indeed, in contrast to BCC, cSCC risk is mostly related to cumulative sun exposure during a lifetime, rather than intermittent and intense sun exposure [17]. Furthermore, it has been widely described that UV exposure, especially during childhood and adolescence, increases the risk of cSCC [189]. The relationship between total lifetime UV exposure and cSCC has been examined in several studies. A strong positive relationship was reported with average annual UV exposure [190], and Rosso et al. described the risk of cSCC as a function of total time spent outdoors [191]. These authors observed a statistically significant increase of risk of cSCC with increasing sun exposure beyond a threshold of 70,000 cumulated hours of exposure in a lifetime [191]. Furthermore, it was reported that outdoor work led to a significantly increased risk of cSCC (OR 1.6 for more than 54,000 cumulated hours of exposure in a lifetime) [191]. Indeed, it has been widely reported that occupational UV exposure is a pivotal risk factor for cSCC, while nonoccupational UV exposure was not reported as significantly related to the development of cSCC [189,192]. However, it must be highlighted that in these studies the UV exposure doses were assessed but not the biological effects of UV exposure on the skin. Indeed, the effects of UV radiation are modified by intrinsic factors, such as Fitzpatrick skin types, age, sex, but also by extrinsic factors like UV protection [189,192].

In experimental and prospective studies, it has been highlighted that regular use of sunscreen reduces the incidence of cSCC [193,194]. The effect of daily sunscreen application in reducing the squamous-cell tumorigenesis was observed by Green et al. [194]. Furthermore, it has been found out that sunscreens prevent UV-induced immune suppression of contact hypersensitivity in mice [195] and can reduce depletion of Langerhans cells [196].

Prophylactic treatments should be considered in patients with a clinical history of multiple cSCC or who show several AKs or a diffuse field cancerization. A prevention with oral retinoids, such as acitretin and isotretinoin, was evaluated in perspective studies in high-risk patients (such as psoriatic patients previously treated with PUVA, patients affected by xeroderma pigmentosum (XP), and solid organ transplant recipients (SOTRs)), showing a significant reduction of development of cSCC [197,198,199,200]. In a nested cohort study on psoriatic patients treated with PUVA, it was reported that 25 mg/d of oral aromatic retinoid reduced risk by about one-fourth. However, this benefit was limited to the period of retinoid intake [197]. In a three-year controlled prospective study, Kraemer et al. reported an average reduction in skin cancers of 63% in XP patients on isotretinoin (2 mg per kilogram of body weight) [199]. However, after discontinuation of the drug, the tumor frequency increased again up to 19-fold (mean value = 8.5-fold) compared to frequency during treatment [199]. Acitretin was also reported as effective in reducing the incidence of cSCC in renal allograft recipients in a double blind, placebo-controlled study [198] and in a prospective study over a five-year period [200]. These results were also confirmed in a prospective, open randomized crossover trial conducted on renal allograft recipients showing the effectiveness of acitretin in preventing cSCC in patients with a previous history of NMSC [201]. However, Chen et al. reported in a recent systematic review that tolerability is the most limiting factor [202]. Furthermore, it has been found that a regular supplement of nicotinamide for one year reduced the rate of new cSCC by 30%, but only during the intake [203,204,205,206]. 

Nonsteroidal anti-inflammatory (NSAIDS) intake was also associated with reduced risk of cSCC in a systematic review and meta-analyses, with the strongest risk reduction observed in current users of COXIB [207,208,209,210].

## Figures and Tables

**Figure 1 biomedicines-09-00171-f001:**
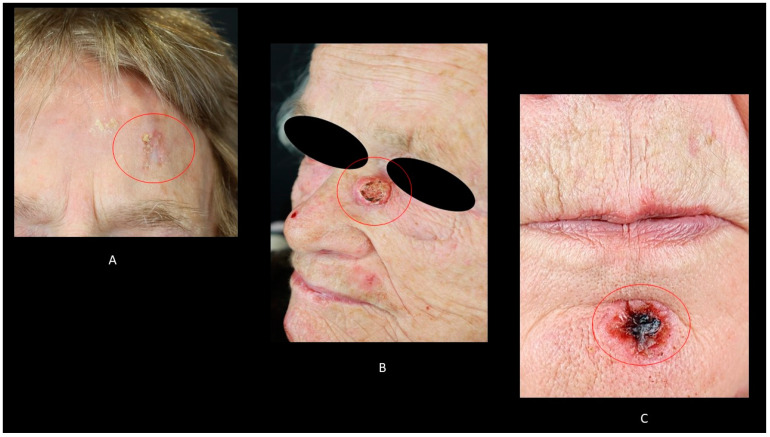
(**A**): Bowen’s disease of the forehead; (**B**): keratoacanthoma of the nose; (**C**): ulcerated squamous cell carcinoma on the chin.

**Figure 2 biomedicines-09-00171-f002:**
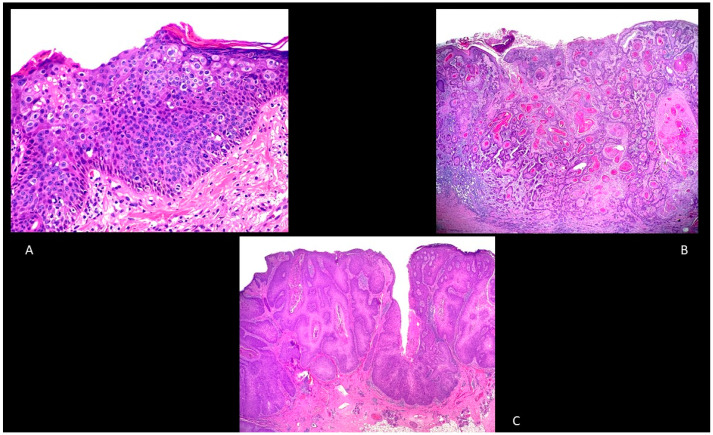
(**A**)**:** Bowen’s disease. Hematoxylin and Eosin, 100× (**B**): Well-differentiated, invasive squamous cell carcinoma, hematoxylin and eosin, 20×. (**C**): Verrucous squamous cell carcinoma, hematoxylin and eosin, 20×.

**Figure 3 biomedicines-09-00171-f003:**
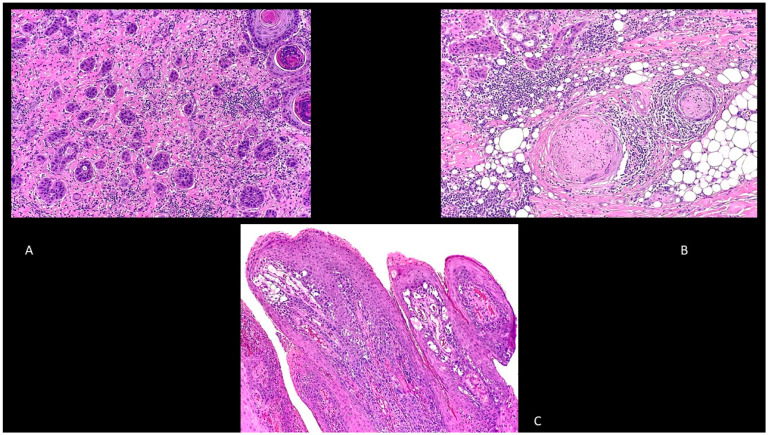
(**A**): Adenosquamous squamous cell carcinoma. Hematoxylin and eosin, 200×. (**B**): Perineural squamous cell carcinoma. Hematoxylin and eosin, 100×. (**C**): Papillary squamous cell carcinoma. Hematoxylin and eosin, 40×.

**Figure 4 biomedicines-09-00171-f004:**
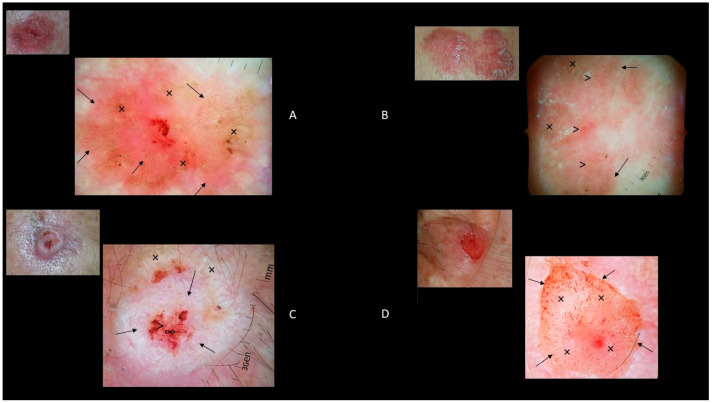
(**A**): Clinical image of a 12 × 8 mm nonpigmented Bowen’s disease of the chest with dermoscopic features of dotted and glomerular vessels (↑) and scaly white-to-yellow surfaces (×). (**B**): Clinical image of a 30 × 12 mm pigmented Bowen’s disease of the leg with dermoscopic features of brown to gray globules/dots (↑) and structureless pigmentation (×) and dotted and glomerular vessels (>). (**C**): Clinical image of a 14 × 11 mm well differentiated cSCC of the scalp with dermoscopic features of keratin/scales (↑), blood spots (>), white structureless areas (×), and ulcerations (∞). (**D**) Clinical image of a 7 × 6 mm poorly differentiated cSCC of the ear with dermoscopic features of hairpin and linear-irregular vessels (↑) and ulcerations (×).

**Figure 5 biomedicines-09-00171-f005:**
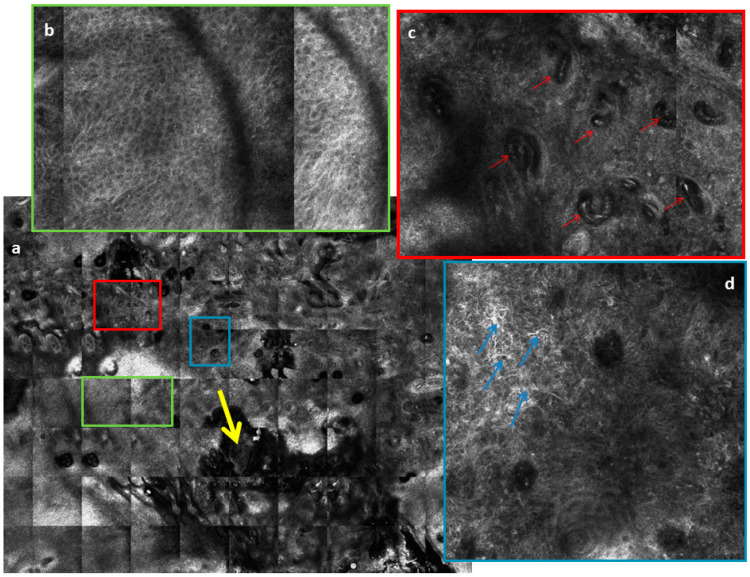
(**a**) Reflectance confocal microscopy (RCM) mosaic of a well differentiated cSCC (8 × 8mm). Yellow arrow shows a scale crust and ulceration in the lower part of the figure. (**b**) Green rectangle: a magnification of an RCM image (1 × 1.5 mm) at the epidermal layer shows atypical honeycomb pattern, which is characterized by thickened and broadened keratinocytes outlines of varying size and shape. (**c**) Red rectangle: a magnification of an RCM mosaic (1 × 1.5 mm) at the dermoepidermal junction layer shows round or coiled vessels (red arrows) that run through the dermal papillae perpendicularly to the lesion surface. (**d**) Blue rectangle: a magnification of an RCM mosaic (1 × 1 mm) at the spinous-granular layer shows a disarranged pattern characterized by architectural disarray and the presence of dendritic (blue arrows) and plump bright cells which represent pigmented keratinocytes and melanocytes infiltrating the epidermis.

**Figure 6 biomedicines-09-00171-f006:**
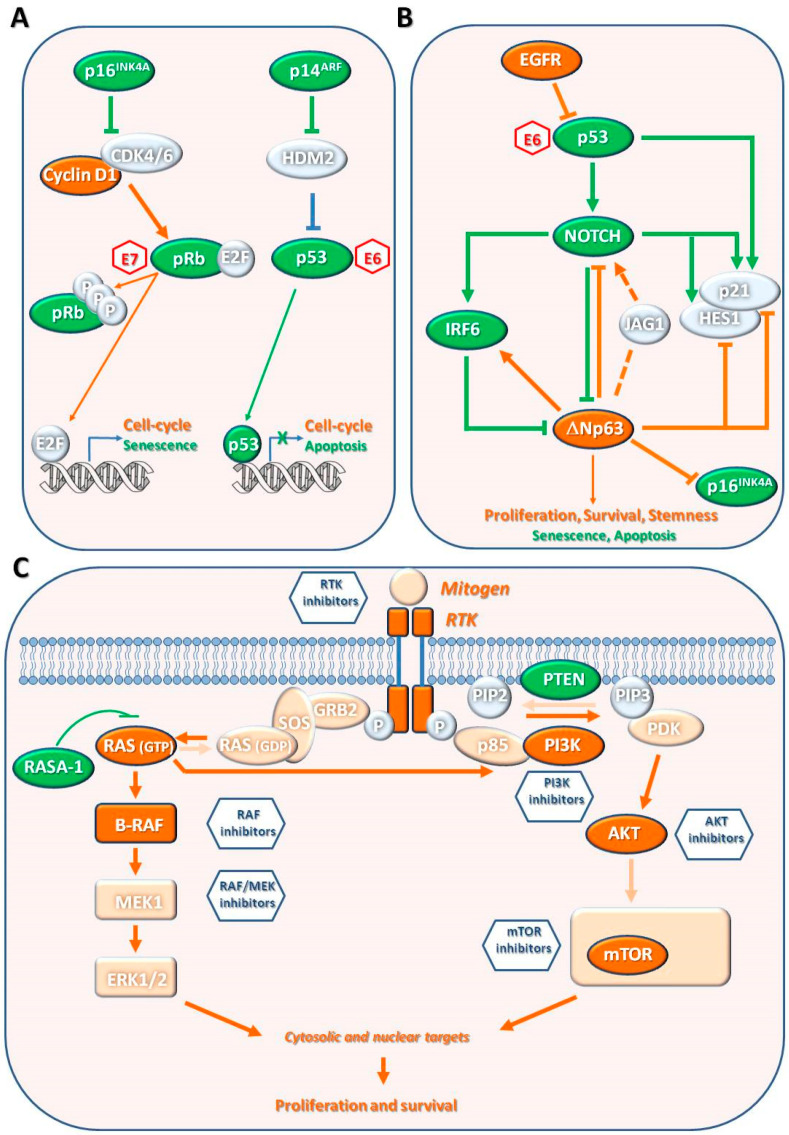
Pathways involved in cutaneous squamous cell carcinoma (cSCC) pathogenesis. Molecular alterations, which drive cSCC development, have been identified in pathways involved in cell cycle regulation, apoptosis, senescence, differentiation, and mitogenic/survival. (**A**) The tumor suppressor genes p16INK4A and p14ARF control retinoblastoma (pRb) and p53 pathways, respectively. Their loss of function promotes cell cycle counteracting senescence or apoptosis. Aberrant activation of E2F transcription can also be due to cyclin D activation or pRb expression loss. pRb and p53 inactivation is also mediated by E6 and E7 (red hexagons) human papilloma virus (HPV) proteins. (**B**) EGF-R aberrant activation, p53 inactivation, or NOTCH gene mutations inactivate the NOTCH pathway. Inactivation of NOTCH abolishes the direct or IRF6-mediated suppression of ΔNp63, favoring proliferation, survival, and stemness. NOTCH inactivation also counteracts senescence and apoptosis mediated by its targets (HES1 and p21). Moreover, ΔNp63 upregulation represses the expression of HES1, p21, and p16INK4A. (**C**) RAS-RAF-MEK-ERK and PI3K/AKT/mTOR pathways share the up-stream proteins, such as tyrosine kinase receptors (RTK) and RAS. Activating mutations in RTK, RAS or inactivation of negative regulator RASA1 promotes cell proliferation and survival through constitutive activation of both pathways. Aberrant activation of these pathways can also derive by B-RAF or PI3K/AKT activation, or Phosphatase and tensin homolog (PTEN) inactivation. The RTKs and the downstream pathways can be targeted with several drugs (blue hexagons) to inhibit cSCC progression. However, both pathways can be activated by RAS mutations, present in photodamaged skin, as part of a compensatory mechanism that could drive resistance to therapeutic targeting strategies. Proteins with commonly accepted tumor promoting and suppressing functions are highlighted in orange and green, respectively. Activated or downregulated processes (circles, squares and arrows) are highlighted in dark orange and green, respectively. Block and dash arrows indicate direct or indirect interactions, respectively.

**Table 1 biomedicines-09-00171-t001:** Clinical and pathological risk factors for recurrence [38].

Clinical Risk Factors for Recurrence	
Size and location of lesion *	≥20 mm on area L ≥10 mm on area M ≥6 mm on area H
Undefined borders	
Recurrent lesion	
SOTRs **	
Previous radiotherapy or site of chronic inflammation	
Rapidly growing lesion	
Neurological symptoms	
**Pathological risk factors for recurrence**	
Not clearly differentiated lesion	
Adenoid, adenosquamous, or desmoplastic subtypes	
Clark level ≥IV	
Modified Breslow thickness ≥4 mm	
Perineural and/or vascular involvement	

* Area H: high risk of recurrence mask areas of face, ears, genitalia, hands, and feet; area M: middle risk of recurrence, cheeks, forehead, neck, and scalp; area L: low risk of recurrence, trunk and extremities. ** SOTRs, solid organ transplant recipients.

**Table 2 biomedicines-09-00171-t002:** Dermoscopic structures of cutaneous squamous cell carcinoma (cSCC) [44].

Nonpigmented Bowen’s Disease	Pigmented Bowen’s Disease	Invasive Squamous Cell Carcinoma
Dotted and glomerular vessels	Brown to gray globules/dots	Keratin/scales
Scaly white-to-yellow surfaces	Structureless pigmentation	Blood spots
	Brown structureless areas	White circles
	Double-edge sign	White structureless areas
		Hairpin and linear-irregular vessels
		Perivascular white halos
		Ulceration

## Data Availability

Not applicable.

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
