# Peer review of "Cutaneous Squamous Cell Carcinoma: From Pathophysiology to Novel Therapeutic Approaches"

_biomedicines, 2021, doi:10.3390/biomedicines9020171_

Round 1

Reviewer 1 Report

The authors submitted the review article on cutaneous squamous cell carcinoma (cSCC). It is not the kind of review that presents new concepts or perspectives on the reviewed subject but instead presents a comprehensive summary of the current knowledge on cSCC, including clinical and histological characteristics, subtypes, treatment options, risk factors and preventions, and molecular mechanisms underlying cSCC. The manuscript generally is well-written and supported by adequate references.

The review is very adequate to the special issue “Skin Cancer: From Pathophysiology to Novel Therapeutic Approaches 2.0”.

Please find below detailed comments that may help to improve the article.

  1. The figures should be better arranged and described. (i) The morphological (Figures 1-3) and histological (Figures 4-9) pictures of cSCC should be presented as panels of one or two figures. Same about clinical pictures (Figures 10-13). (ii) I would also suggest to indicate on the figures, features characteristic to particular subtypes, described in the text. (iii) Figure panels should be consequently labeled either with capital (A, B, ...) or lowercase (a, b, ...) letters. (iv) Figure 15, some fonts are difficult to read.
  2. Figure 15 legend is strongly redundant with the text. Redundancies should be removed.
  3. The BD and other abbreviations should be used consequently through the manuscript, including figure legends and Tables. The abbreviations should not be explained more than once. Please double-check the manuscript.
  4. Gene IDs should be italicized.
  5. Line 364; is p53; should be TP53. Same in line 400.
  6. Lines 397-398: I think, it should be (7-48% of samples). Please double-check.
  7. Delta symbols in lines 477-479 should be corrected or explained. Also in another part of the manuscript.

Author Response

Dear Editor,

Thank you for giving us the opportunity to revise and improve our manuscript.

Please find our point-by-point response to the reviewers’ comments/suggestions.

All changes are highlighted in the revised manuscript.

We have followed the suggestions of the reviewers, and we now hope that our manuscript will be suitable for publication in Your journal.

Best regards,

Reviewer #1

Comment 1: The figures should be better arranged and described. (i) The morphological (Figures 1-3) and histological (Figures 4-9) pictures of cSCC should be presented as panels of one or two figures. Same about clinical pictures (Figures 10-13). (ii) I would also suggest to indicate on the figures, features characteristic to particular subtypes, described in the text. (iii) Figure panels should be consequently labeled either with capital (A, B, ...) or lowercase (a, b, ...) letters. (iv)

Figure 15, some fonts are difficult to read.

Response: Figure 1 to 13 have been modified and we created 4 panels with labeled letters. Characteristic features have been indicated in some figures. The fonts of Figure 15, now 6, have been modified.

Comment 2: Figure 15 legend is strongly redundant with the text. Redundancies should be removed.

Response: Figure 15 legend, now 6, has been modified and shorten.

Comment 3: The BD and other abbreviations should be used consequently through the manuscript, including figure legends and Tables. The abbreviations should not be explained more than once. Please double-check the manuscript.

Response: All abbreviations have been checked and corrected

Comment 4: Gene IDs should be italicized.

Response: Gene IDs have been italicized

Comment 5: Line 364; is p53; should be TP53. Same in line 400.

Response: Modified

Comments 6: Lines 397-398: I think, it should be (7-48% of samples). Please double-check.

Response: Modified

Comment 7: Delta symbols in lines 477-479 should be corrected or explained. Also in another part of the manuscript.

Response: Delta symbols have been corrected

Reviewer 2 Report

The authors are to be commended for this excellent review. Although the paper is well written, comprehensive, heavily up-dated, easy to read, a guideline for the clinicians and researchers involved in the study and treatment of cutaneous squamous cell carcinoma, a few minor suggestions should be addressed in order to increase the impact of this paper.

Not all the listed keywords are detected in the abstract section. Please clarify.

I would suggest in order to follow a didactic path, that the section on Clinical Features (line 127) should be listed following the section on Pathogenesis (line 340).

Author Response

Dear Editor,

Thank you for giving us the opportunity to revise and improve our manuscript.

Please find our point-by-point response to the reviewers’ comments/suggestions.

All changes are highlighted in the revised manuscript.

We have followed the suggestions of the reviewers, and we now hope that our manuscript will be suitable for publication in Your journal.

Best regards,

Reviewer #2

Comment 1: Not all the listed keywords are detected in the abstract section. Please clarify.

Response: Abstract and keywords have been modified and now all keywords are listed in the                 abstract

Comment 1: I would suggest in order to follow a didactic path, that the section on Clinical Features (line        127) should be listed following the section on Pathogenesis (line 340).

Response: We thank the reviewer for his/her comment. We decided to put the pathogenesis section subsequently to clinical features and before the therapy because this section (pathogenesis) explained important features linked to the treatment, mainly with systemic treatment (e.g., immunotherapy). This order has been even utilized in the review regarding “basal cell cariconoma” that we recently published in the same journal (“Biomedicines”).
